# TAILORING LANGUAGE GENERATION MODELS UNDER TOTAL VARIATION DISTANCE

**Haozhe Ji**[1], **Pei Ke**[1], **Zhipeng Hu**[2], **Rongsheng Zhang**[2], **Minlie Huang**[1]*

[1]Dept. of Comp. Sci. & Tech., State Key Lab of Intelligent Tech. & Sys.
[1]BNRist Center, Tsinghua University, Beijing 100084, China
[2]Fuxi AI Lab, NetEase Inc., China
`jhz20@mails.tsinghua.edu.cn, kepei1106@outlook.com`
`{zphu, zhangrongsheng}@corp.netease.com, aihuang@mail.tsinghua.edu.cn`

## ABSTRACT

The standard paradigm of neural language generation adopts maximum likelihood estimation (MLE) as the optimizing method. From a distributional view, MLE in fact minimizes the Kullback-Leibler divergence (KLD) between the distribution of the real data and that of the model. However, this approach forces the model to distribute non-zero (sometimes large) probability mass to all training samples regardless of their quality. Moreover, in the attempt to cover the low-probability regions in the data distribution, the model systematically overestimates the probability of corrupted text sequences, which we conjecture is one of the main reasons for text degeneration during autoregressive decoding. To remedy this problem, we leverage the total variation distance (TVD) with its robustness to outliers, and develop practical bounds to apply it to language generation. Then, we introduce the TaiLr[1] objective that balances the tradeoff of estimating TVD. Intuitively, TaiLr downweights real data samples that have low model probabilities with tunable penalization intensity. Experimental results show that our method alleviates the overestimation of degenerated sequences without sacrificing diversity and improves generation quality on a wide range of text generation tasks.[2]

## 1 INTRODUCTION

The dominant approach to train language generation models is to maximize the likelihood of text samples in training data. With the development of pre-training techniques, the quality of texts generated by current models has been improved by a large margin (Radford et al., 2019; Brown et al., 2020). However, the text degeneration phenomena, e.g., repetitions (Holtzman et al., 2020; Welleck et al., 2020), incoherence (Guan et al., 2021; Ji & Huang, 2021), and other ill-formed generation results sampled from the noisy long tail (Dou et al., 2022; LeBrun et al., 2022), are still widely observed in large pre-trained models. These results indicate that using MLE as the optimizing method has theoretical limitations that are hard to be compensated by increasing the model size.

Given the real data distribution $p(\boldsymbol{x})$ and the model distribution $q(\boldsymbol{x})$ defined by a learned generation model, we can view MLE as minimizing the KLD between $p(\boldsymbol{x})$ and $q(\boldsymbol{x})$. However, minimizing $D_{\mathrm{KL}}(p, q)$ will lead to a *zero-avoiding* solution of $q(\boldsymbol{x})$ that spreads itself to cover all the modes in the real data (Minka, 2005; Malinin & Gales, 2019). As the model is forced to take into account all the modes regardless of their quality and saliency, this behavior could deteriorate the overall generation quality when (i) the data inherently exhibits too many variations, e.g., in open-ended generation, the model often over-presents unrelated words in the unreliable long tail of its distribution (Holtzman et al., 2020). (ii) the data contains flawed or noisy references, e.g., hallucination and missing contents in text summarization (Zhao et al., 2020) degrade the generation quality of the model.

In language generation, the attempt to cover all the non-zero probability regions in the data distribution would lead to a problem directly related to text degeneration, which we term as *data void*

---

*Corresponding Author.
[1]Pronounced as "tailor".
[2]Code is available at `https://github.com/thu-coai/TaiLr`.

*overestimation*. Concretely, the model assigns considerably more probability mass than it should to the *void* of the real data distribution, where degenerated text sequences lie. An intuitive illustration is shown in Figure 1 where KLD pushes the model to place large mass on the zero-probability region of the target distribution to cover the minor mass portion on the right. These degenerated texts include random word sequences and partially corrupted texts that have high lexical overlap with the real texts. Therefore, during free-run generation, the model is likely to trap into the void regions and produce "over-generalized" text samples that are unlike the training data (Huszar, 2015).

In this work, we start with a robust alternative to KL divergence, i.e., the total variation distance (TVD). TVD is known to be robust to outliers in the data (Beran, 1977; Knoblauch & Vomfell, 2020), as it measures the absolute difference between two probability distributions averaging at each point. In §2.2, we show that TVD allows the model to place zero probability to low-quality training samples and prevent overestimation of the data void region through gradient analysis. Though appealing, TVD cannot be directly applied to text generation because (i) TVD measures the distance at the sequence level while we desire a token-level criterion for autoregressive generation models, (ii) we only have samples from the data distribution, whereas calculating TVD demands the real data probability $p(\boldsymbol{x})$ of the training sample $\boldsymbol{x}$. We overcome these two issues by (i) developing an upper bound on the sequence-level TVD with its token-level factorization (§3.1), and (ii) introducing a proxy distribution (§3.2) that handles the bias-variance tradeoff during estimating TVD (§3.3). Finally, we derive the **T**otal **Va**riation Gu**i**ded **L**anguage Gene**r**ation (TaiLr) objective by leveraging access to the non-zero gradient of TVD to guide the model. Intuitively, TaiLr weights the log-likelihood of a text sequence at each position according to the model probability and uses a tunable hyperparameter to control the penalization intensity.

We first conduct experiments on synthetic data to show that TaiLr achieves better generation quality without sacrificing diversity and reduces the overestimation of degenerated texts compared to MLE. Further experiments on real data demonstrate that the proposed method outperforms existing methods that modify MLE at different aspects on a wide range of language generation tasks, including machine translation, text summarization, and long text generation.

## 2 BACKGROUND AND MOTIVATION

We consider natural language generation tasks where a conditional generation model $p_\theta(\boldsymbol{y}|\boldsymbol{x})$ parametrized by $\theta$ is required to generate the target text sequence $\boldsymbol{y} = (y_1, \cdots, y_T)$ given the context $\boldsymbol{x}$. Let $p_o(\boldsymbol{y}|\boldsymbol{x})$ denote the real data distribution, MLE training is equivalent to minimizing the KL divergence between $p_o$ and $p_\theta$:

$$D_{\mathrm{KL}}(p_o, p_\theta) = -\mathbb{E}_{\boldsymbol{y} \sim p_o}\left[\sum_{t=1}^{T} \log p_\theta(y_t|\boldsymbol{y}_{<t}, \boldsymbol{x})\right] - H(p_o), \tag{1}$$

where the generation probability is factorized into the product of conditional token probabilities given the prefix $\boldsymbol{y}_{<t}$ and the context $\boldsymbol{x}$: $p_\theta(\boldsymbol{y}|\boldsymbol{x}) = \prod_{t=1}^{T} p_\theta(y_t|\boldsymbol{y}_{<t}, \boldsymbol{x})$. The first term pushes the model to minimize the negative log-likelihood (NLL) of the training data. The second term is a constant with respect to $\theta$ and therefore is commonly ignored in MLE.

Despite its simplicity and practical benefits for optimization, MLE is known to suffer from a mismatch to the evaluation metric (Pang & He, 2021) and brittleness to noise in the training data (Kang & Hashimoto, 2020). Motivated by the literature in probability metrics, we draw attention to total variation distance (TVD) as a naturally robust alternative to KLD. We present the definitions of TVD (Van Handel, 2014) between the data distribution $p_o$ and the model distribution $p_\theta$ given the context $\boldsymbol{x}$:

$$D_{\mathrm{TV}}(p_o, p_\theta) = \frac{1}{2}\sum_{\boldsymbol{y} \in \mathcal{Y}} \left| p_o(\boldsymbol{y}|\boldsymbol{x}) - p_\theta(\boldsymbol{y}|\boldsymbol{x}) \right| \tag{2a}$$

$$= 1 - \sum_{\boldsymbol{y} \in \mathcal{Y}} \min\left( p_o(\boldsymbol{y}|\boldsymbol{x}), p_\theta(\boldsymbol{y}|\boldsymbol{x}) \right), \tag{2b}$$

where $\mathcal{Y}$ is the space of all possible text sequences. Intuitively, TVD measures the average of the absolute difference between $p_o(\boldsymbol{y}|\boldsymbol{x})$ and $p_\theta(\boldsymbol{y}|\boldsymbol{x})$ on all possible text sequence $\boldsymbol{y} \in \mathcal{Y}$. Therefore

the model learns to properly allocate its probability mass to best describe the major part of the data distribution and ignore outliers. TVD is also correlated with the *distinguishability* of samples generated by the model, which is shown to be a balanced criterion that takes both quality and diversity into account (Hashimoto et al., 2019). Existing work proposed to optimize distinguishability in an generative adversarial manner (Goodfellow, 2015; Caccia et al., 2020) while Kang & Hashimoto (2020) argued that minimizing its heuristic surrogate via loss truncation is better in practice. Additional related work is provided in Appendix B. In this work, we first analyze the property of TVD and seek to directly minimize TVD or at least its upper bound in the task of natural language generation.

## 2.1 A Toy Experiment and Its Implications

We first present a toy experiment to illustrate the behavioral difference of KLD and TVD when countering imperfect data, where a single Gaussian model is required to fit a mixture of two Gaussians. As shown in Figure 1, minimizing KLD forces the model to learn a flat distribution that spans itself to cover all the non-zero probability regions, which causes underfitting of the major part of the target probability mass. Furthermore, the model places considerably high probability mass to the *void region* in the target distribution which does not correspond to real samples. On the other hand, TVD focuses on the major target mass without overestimating degenerated samples that are unlikely under the target distribution.

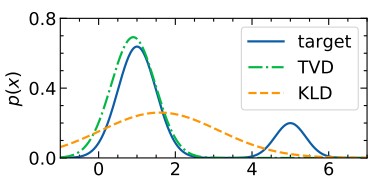

Figure 1: Results of the toy experiment: KLD is sensitive to outliers while TVD is more robust.

In language generation, this scenario is realistic and pervasive. For many language geneartion tasks, it is hard to circumvent noisy or invalid references during the data collection process, e.g., hallucination in text summarization (Zhao et al., 2020) and image captioning (Xiao & Wang, 2021). For applications like open-ended generation, existing autoregressive models pre-trained on large corpus are still reported to over-present the artifacts in the noisy long tail (Holtzman et al., 2020).

## 2.2 Gradient Analysis

To better understand the reason behind the behaviorial difference of KLD and TVD in optimization, we analyze their gradients with respect to the model parameter $\theta$. Given a context-target text pair $(\boldsymbol{x}^*, \boldsymbol{y}^*)$ sampled from the data distribution $p_o$, we approximate the gradient of KLD with respect to $\theta$ using Monte-Carlo sampling[3]:

$$\nabla_\theta D_{\mathrm{KL}}(p_o, p_\theta) \approx -p_\theta(\boldsymbol{y}^*|\boldsymbol{x}^*)^{-1} \nabla_\theta p_\theta(\boldsymbol{y}^*|\boldsymbol{x}^*). \tag{3}$$

The result is the negative gradient of the model probability weighted by the reciprocal of the model probability on this sample. Intuitively, when a low-quality context-target pair is sampled, the model will be affected by this sample and shift the distribution towards it. If $p_\theta(\boldsymbol{y}^*|\boldsymbol{x}^*) \approx 0$, the norm of the gradient will become very large, which leads to a huge step of parameter update towards that noisy direction. This explains the phenomena illustrated in §2.1, where KLD pushes the model to cover all the training samples resulting in an unfocused and flat distribution.

For comparison, we calculate the gradient of TVD with respect to $\theta$ using equation (2b). The derivation details are provided in the Appendix A.1.

$$\nabla_\theta D_{\mathrm{TV}}(p_o, p_\theta) \approx \begin{cases} -p_o(\boldsymbol{y}^*|\boldsymbol{x}^*)^{-1} \nabla_\theta p_\theta(\boldsymbol{y}^*|\boldsymbol{x}^*), & p_\theta(\boldsymbol{y}^*|\boldsymbol{x}^*) < p_o(\boldsymbol{y}^*|\boldsymbol{x}^*) \\ 0, & p_\theta(\boldsymbol{y}^*|\boldsymbol{x}^*) \geq p_o(\boldsymbol{y}^*|\boldsymbol{x}^*), \end{cases} \tag{4}$$

where the result switches between a non-zero gradient term and 0 by comparing the model probability and the real data probability. When the model probability exceeds the real data probability (**overestimation**), the gradient becomes 0 to prevent the model from fitting dubious data points. When the model predicts a probability lower than the real probability of the sample (**underestimation**), the weight is the reciprocal of the real probability of the sample, which has a smaller norm than equation (3). This means that the update towards noisy directions is more conservative, and the model is allowed to assign 0 probability to those low-quality training samples.

---

[3]For clarity, we only use one sample per batch in this analysis, and the result still holds for large batch size.

## 3 METHODOLOGY

Despite the attractive attribute of TVD, we still face several challenges to apply TVD to natural language generation. First, TVD measures the difference of the sequence-level probability. For autoregressive language generation models, it is typical to use a token-level criterion to supervise the factorized model probability. Although the sequence-level objective can also be adopted as a reward function using policy gradient (Williams, 1992; Sutton et al., 1999), this approach is shown to suffer from a high variance and sparse rewards. Second, calculating TVD requires the real data probability $p_o(\boldsymbol{y}|\boldsymbol{x})$ of the sample $\boldsymbol{y}$ to be known. One straightforward solution is to train a classifier that estimates the density ratio between $p_o(\boldsymbol{y}|\boldsymbol{x})$ and $p_\theta(\boldsymbol{y}|\boldsymbol{x})$ (Song et al., 2020). However, the density ratio estimator would introduce undetermined biases due to miscalibration (Grover et al., 2019). In this work, we tackle these challenges by developing practical upper bounds on TVD, and derive a sampling-based learning criterion which can directly substitute for the MLE objective in practice.

### 3.1 TOKEN-LEVEL FACTORIZATION

As KLD has the nice property of factorizing the sequence-level loss into summation of the token-level loss conditioned on the prefix as illustrated in equation (1), we wonder if TVD also has this property. We first write the autoregressive factorization of the data probability as $p_o(\boldsymbol{y}|\boldsymbol{x}) = \prod_{t=1}^{T} p_o(y_t|\boldsymbol{y}_{<t}, \boldsymbol{x})$. For simplicity, we use $p_o^{<t}(y_t)$ and $p_\theta^{<t}(y_t)$ to denote $p_o(y_t|\boldsymbol{y}_{<t}, \boldsymbol{x})$ and $p_\theta(y_t|\boldsymbol{y}_{<t}, \boldsymbol{x})$ respectively. Then we have the following proposition that manifests the relationship between the sequence-level objective and its token-level factorization.

**Proposition 1.** *Given $p_o(\boldsymbol{y}|\boldsymbol{x}) = \prod_{t=1}^{T} p_o^{<t}(y_t)$ and $p_\theta(\boldsymbol{y}|\boldsymbol{x}) = \prod_{t=1}^{T} p_\theta^{<t}(y_t)$, then the following condition holds:*

$$D_{\mathrm{TV}}(p_o, p_\theta) \leq \mathbb{E}_{\boldsymbol{y} \sim p_o} \left[ \sum_{t=1}^{T} D_{\mathrm{TV}}(p_o^{<t}, p_\theta^{<t}) \right]. \tag{5}$$

The condition follows from applying triangle inequality (Hein & Bousquet, 2005) to the right hand side of equation (2a). The complete proof is provided in Appendix A.2. This result indicates that minimizing the expected sum of the TVD on token-level probabilities is equivalent to minimizing the upper bound of the TVD on their products where the bound becomes tight as $p_\theta$ approaches $p_o$. Therefore, we are guaranteed to train the model using the MLE fashion that calculates the loss at each position given the prefix of the target sequence.

### 3.2 ESTIMATION WITH PROXY DISTRIBUTION

Another difficulty of directly applying TVD to train language generation model is the explicit demand of the data probability distribution $p_o$ while we only have a finite number of samples drawn from it. In contrast to using an additional density ratio estimation model that is both hard to train and potentially biased with undetermined deviation, we try to estimate the target using a *proxy probability distribution* and analyze the estimation error.

We start by considering the one-hot distribution $e^{(w)}$ where only the $w$-th index is 1 and others are 0. $w$ is the target token sampled from the conditional oracle probability $p_o^{<t}(\cdot)$. It is easy to see that the expectation of the one-hot distribution is exactly the oracle probability distribution:

$$\mathbb{E}_{w \sim p_o^{<t}} \left[ e^{(w)} \right] = p_o^{<t}. \tag{6}$$

Then we use $e^{(w)}$ to substitute the oracle probability $p_o^{<t}$ in TVD and present the following proposition which states that the expectation of this estimation serves as an upper bound of the original TVD between the oracle distribution and the model distribution.

**Proposition 2.** *Given $w \sim p_o^{<t}$ and the one-hot distribution $e^{(w)}$, then the following condition holds:*

$$D_{\mathrm{TV}}(p_o^{<t}, p_\theta^{<t}) \leq \mathbb{E}_{w \sim p_o^{<t}} \left[ D_{\mathrm{TV}}(e^{(w)}, p_\theta^{<t}) \right]. \tag{7}$$

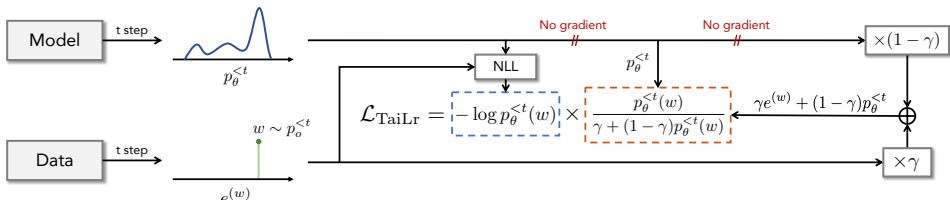

Figure 2: Computational graph of the TaiLr objective where the log-likelihood is weighted position-wisely.

The proof utilizes the Jensen inequality and the convexity of the TVD. The full proof is presented in Appendix A.3. The proposition states that minimizing the TVD between the model distribution and the one-hot distribution *on average* leads to the minimization of the TVD between the model distribution and the oracle distribution. By introducing an *unbiased* estimation of the unknown oracle distribution, we sidestep the need of density ratio estimation and derive a practical upper bound using Monte-Carlo sampling from the oracle distribution.

### 3.3 THE BIAS-VARIANCE TRADEOFF

However, using the one-hot distribution as an approximation in practice sometimes leads to high estimation variance. For example, in some applications where the real data distribution has a high entropy, using the one-hot proxy can hardly cover the diverse candidates. Therefore we consider a general form of the proxy distribution $\hat{p}^{(w)}$ where $w$ is the target token. We denote the expectation of the general proxy distribution as $\hat{p}^{<t} = \mathbb{E}_{w \sim p_o^{<t}}\left[\hat{p}^{(w)}\right]$. Then we show that the upper bound of the estimation error can be decomposed into a bias term and a variance term in the following:

$$\text{Error}_{\hat{p}^{(w)}} \leq \underbrace{D_{\text{TV}}(\hat{p}^{<t}, p_o^{<t})}_{\text{Bias}} + \underbrace{\mathbb{E}_{w \sim p_o^{<t}}\left[D_{\text{TV}}(\hat{p}^{(w)}, \hat{p}^{<t})\right]}_{\text{Variance}}, \tag{8}$$

where $\text{Error}_{\hat{p}^{(w)}}$ is defined as the difference of the practical estimation $\mathbb{E}_{w \sim p_o^{<t}}\left[D_{\text{TV}}(\hat{p}^{(w)}, p_\theta^{<t})\right]$ and the ideal target $D_{\text{TV}}(p_o^{<t}, p_\theta^{<t})$. The derivation applies triangle inequality to bound the error term (detailed derivation can be found in Appendix A.4). Specifically, we consider the one-hot distribution as an example: it has zero estimation bias (equation (6)). However we show in Appendix A.5 that its variance equals to $2H_\alpha(p_o^{<t})$ when $\alpha = 2$, where $H_\alpha$ is the Tsallis $\alpha$-entropy (Tsallis, 1988). Therefore, the one-hot proxy suffers from a large variance when the entropy of $p_o^{<t}$ is high.

In order to handle the bias-variance tradeoff, we consider a $\gamma$-mixture proxy distribution that interpolates the one-hot distribution and the model distribution with $\gamma$: $\hat{p}^{(w)} = \gamma e^{(w)} + (1 - \gamma)p_\theta^{<t}$. Below we show the bias and variance in equation (8) using this mixture proxy distribution:

$$\text{Bias} = (1 - \gamma) \cdot D_{\text{TV}}(p_\theta^{<t}, p_o^{<t}), \quad \text{Variance} = \gamma \cdot \mathbb{E}_{w \sim p_o^{<t}}\left[D_{\text{TV}}(e^{(w)}, p_o^{<t})\right]. \tag{9}$$

When we tune $\gamma$ from 1 to 0, the proxy distribution smoothly transfers from the unbiased one-hot distribution to a soft distribution, which reduces the variance of the one-hot estimation and stablizes training in the early stage. Although this comes at the cost of an increased estimation bias at the beginning of training, the bias gradually decreases as the model fits the data distribution more accurately when the training goes on.

### 3.4 TOTAL VARIATION GUIDED LANGUAGE GENERATION (TAILR)

Finally, we introduce the TaiLr objective by summarizing the above results. Given the target token $w$, we derive the TVD between the proxy distribution $\hat{p}^{(w)} = \gamma e^{(w)} + (1 - \gamma)p_\theta^{<t}$ and the model distribution $p_\theta^{<t}$ following equation (2b):

$$D_{\text{TV}}(\hat{p}^{(w)}, p_\theta^{<t}) = 1 - \mathbb{E}_{y_t \sim \hat{p}^{(w)}} \min\left(1, \frac{p_\theta^{<t}(y_t)}{\hat{p}^{(w)}(y_t)}\right), \tag{10}$$

where the expectation is approximated by sampling from the proxy distribution using Monte-Carlo sampling. When sampling $y_t \neq w$, the gradient of $D_{\text{TV}}(\hat{p}^{(w)}, p_\theta^{<t})$ is always 0 which is inefficient for optimization. Therefore, we consider the non-zero gradient when $y_t$ is sampled as the target token $w$ to guide the model, i.e., $-\nabla_\theta p_\theta^{<t}(w)/\hat{p}^{(w)}(w)$, and devise the TaiLr objective whose gradient is equivalent to it:

$$\mathcal{L}_{\text{TaiLr}}(w; \theta) = -\left( \frac{p_\theta^{<t}(w)}{\gamma + (1 - \gamma)p_\theta^{<t}(w)} \right) \log p_\theta^{<t}(w), \tag{11}$$

where the weighting factor is detached in the back-propagation and only the log term receives gradient. The equivalence of $\nabla_\theta \mathcal{L}_{\text{TaiLr}}(w; \theta)$ and the non-zero gradient of $D_{\text{TV}}(\hat{p}^{(w)}, p_\theta^{<t})$ can be seen by applying $f(x)\nabla_x \log f(x) = \nabla_x f(x)$. In Figure 2, we show the computational graph of the TaiLr objective. As $\gamma$ switches from 1 to 0, TaiLr is biased from an estimation of TVD to unweighted MLE. Intuitively, TaiLr downweights samples with low probabilities assigned by the model so that the model focuses on modeling the high-quality samples during training and reduces overestimation of degenerated texts during inference. To counter the negative effect of random prediction at the early training stage, we set a threshold as a lower bound of the weighting factor.

## 4 EXPERIMENTS

In the previous sections, we show that the proposed method is a practical estimation of TVD with theoretical guarantees. Next, we demonstrate its empirical performance. First, we conduct a synthetic experiment to investigate the behavior of the model trained by TaiLr and MLE in controlled settings where the underlying oracle distribution is known. Second, we compare TaiLr with other baselines in a more realistic setting where we train generation models with standard architectures or finetune pre-trained models on a wide range of language generation benchmarks. More experimental details which are not included in the following sections are provided in Appendix E.1.

### 4.1 SYNTHETIC EXPERIMENTS

**The synthetic data.** In this subsection, our goal is to test the behavior of TaiLr in the task of text generation. Since we seek to analyze the distributional properties, we sample training data an oracle model whose distribution is known. Instead of using random distributions (Yu et al., 2017; Guo et al., 2018), we follow LeBrun et al. (2022) and train an oracle model on real human texts to generate synthetic data so that the results can better generalize to real data. Specifically, we train a 1-layer LSTM on the texts of the COCO image caption dataset (Lin et al., 2014) without any conditional inputs. We sample 10K synthetic data for training and 5K for validation.

**The model setting.** We train two LSTMs with the same architecture as the oracle model using MLE and TaiLr, which we denoted as $p_{\text{MLE}}$ and $p_{\text{TaiLr}}$, respectively. We train both models for 100 epochs and pick the best checkpoint with the lowest perplexity on the development set. We use random sampling to obtain text samples from the learned generation models.

| Model | PPL$_{oracle}$ ↓ | PPL$_{test}$ ↓ | BLEU-4 ↑ | SelfBLEU-4 ↓ |
|---|---|---|---|---|
| Training data | - | - | **35.40** | 30.83 |
| $p_{\text{MLE}}$ | 31.64 | 22.64 | 27.74 | 33.27 |
| $p_{\text{TaiLr}}$ | **26.91** | **22.42** | 28.36 | **28.91** |

Table 1: Automatic evaluation results of the models trained by MLE and TaiLr. PPL$_{oracle}$ and BLEU assess the generation quality, while PPL$_{test}$ and SelfBLEU emphasize on sample diversity. **Boldface** and underline indicate the highest and the second highest performance respectively.

**Performance evaluation.** To thoroughly evaluate the generation performance of the two models, we follow Yu et al. (2017); Caccia et al. (2020) to evaluate the generation quality with PPL$_{oracle}$ and the coverage of the oracle distribution with PPL$_{test}$. Specifically, PPL$_{oracle}$ is the likelihood of the oracle model calculated on the samples generated by the learned model, while PPL$_{test}$ is the likelihood of the learned model evaluated on the held-out data. We also include BLEU score (Papineni et al., 2002) to calculate the average $n$-gram overlap between the generated sample and the

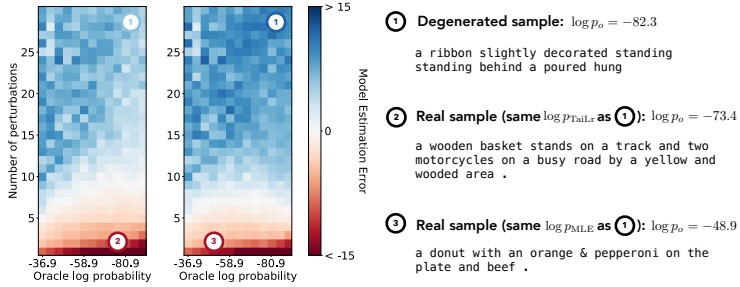
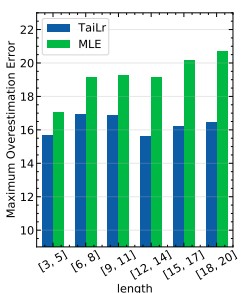

Figure 3: Estimation error map of $p_{\text{TaiLr}}$ (left) and $p_{\text{MLE}}$ (right) on the perturbed dataset. We present examples from different regions to illustrate the estimation behavior of the two models.

Figure 4: Maximum overestimation error varying with lengths.

held-out corpus, and SelfBLEU (Zhu et al., 2018) which computes the average overlap of each generated sample to other samples generated by the model[4]. For evaluation, we use 20K held-out data and sample 20K texts from the two generation models, respectively. As shown in Table 1, TaiLr improves the generation quality by nearly 5 points of $\text{PPL}_{oracle}$ without sacrificing the coverage to the oracle distribution as it achieves similar $\text{PPL}_{test}$ as MLE. We also observe that TaiLr achieves higher BLEU-4 than MLE while lower than the training data. Finally, we show that MLE has the highest SelfBLEU-4 which shows its tendency to over-generalize to unseen samples that may include repeated patterns and degrade the diversity, while TaiLr achieves the lowest SelfBLEU-4. We also report the result using GPT2-Medium as the oracle model in Appendix E.1.1.

**Perturbation evaluation.** Next, we evaluate models' behavior on perturbed sequences and relate text degeneration as an implication of model's overestimation behavior on the pertubed data. To quantify the deviation of the model's estimation from the real data distribution, we define the model estimation error of a sample $x$ as the difference between the sequence-level log probability given by the model and its true log probability, i.e., $\text{Error}(x) = \log p_\theta(x) - \log p_o(x)$. Then we show the construction of the perturbed dataset. Given $x$ sampled from $p_o$, we iteratively apply small perturbations to $x$ so that each lexical change is small. After $N$ perturbations, $x \to x^{(1)} \to \cdots \to x^{(N)}$ smoothly transfers a data point into a perturbed sample. We propose the following perturbations that highlight the widely observed text degeneracy patterns in generation: (1) **Repeat** a token in $x$ (repetition, Welleck et al. (2020)). (2) **Delete** the last token in $x$ (oversmoothing, Kulikov et al. (2021)). (3) **Substitute** a token in $x$ with a token from the vocabulary (incoherence, Holtzman et al. (2020)). We sample 20K samples from $p_o$ and apply $N = 30$ perturbations to each sample.

We first plot the estimation error map of the two models on the perturbed dataset in Figure 3. For each perturbed sample $x^{(i)}$, the oracle log probability $\log p_o(x^{(i)})$ is shown in the x-axis, the number of perturbations $i$ performed is shown in the y-axis, and the estimation error $\text{Error}(x^{(i)})$ is reflected by the shade. From the figures on the left, we first restate LeBrun et al. (2022)'s finding that exisiting models underestimate real samples while overestimating degenerated samples. Next, by comparing the two figures, we observe that as the number of perturbations increases, $p_{\text{TaiLr}}$ alleviates the overestimation phenomenon of $p_{\text{MLE}}$ especially in the long tail. Finally, we draw cases from different regions in the figure to illustrate its implication on text degeneration. We first present a degenerated sample ① on the top-right corner which has low oracle probability. We then present two real data samples ② and ③ that have the same model probability under $p_{\text{TaiLr}}$ and $p_{\text{MLE}}$ as the degenerated one ①. Although ③ is actually more probable than the degenerated one ① in the oracle distribution, MLE cannot distinguish between them leading to degeneracy patterns during generation.

To quantify the overestimation problem of perturbation, we further define the maximum overestimation error over $N$ perturbations as $\max\limits_{i=1,\cdots,N} \text{Error}(x^{(i)})$. To manifest the overestimation problem during generation, we plot the maximum overestimation error averaged on samples grouped with the similar length in Figure 4. Note that the average NLL of these degenerated samples for $p_{\text{MLE}}$ and $p_{\text{TaiLr}}$ is 11.09 and 10.82 respectively. For MLE the overestimation error amplifies as the generation

---

[4]The scripts of both BLEU and SelfBLEU are from `https://github.com/geek-ai/Texygen/blob/master/utils/metrics`

length increases while TaiLr maintains the error nearly at a constant. This result demonstrates that TaiLr alleviates MLE' s tendency to sample degenerated texts with the growth of the generation length by weighting the likelihood at each position of the sequence during training.

**Error accumulation analysis.** Finally, we analyze the error accumulation during autoregressive decoding. We follow Arora et al. (2022) and use the metric ExAccErr that calculates the percentage of *excess* errors due to the discrepency of training (conditioning on contexts sampled from $p_o$) and inference (conditioning on contexts sampled from $p_\theta$), i.e., exposure bias. Detailed definitions borrowed from Arora et al. (2022) are provided in Appendix E.1.2. We found that the excess error of MLE model ($40.1\%$) is substantially higher than the model trained with TaiLr ($8.6\%$), which demonstrates that TaiLr effectively reduces the error accumulation during autoregressive decoding.

## 4.2 REAL-DATA EXPERIMENTS

In this subsection, we describe the empirical evaluation of TaiLr on a wide range of real-world language generation tasks, including: (1) **Machine Translation**: Given a sentence in the source language, the goal is to translate it into the target language. (2) **Text summarization**: Given a passage, the goal is to generate a short sentence that summarizes the main point of the passage. (3) **Long text generation**: Given a title, the goal is to generate a coherent long passage that conforms with the title. Statistics and sources of all datasets used in experiments are provided in Appendix D.

Apart from MLE, we also consider the following typical baselines that proposed new training objectives beyond MLE: (1) **Unlikelihood training** (Welleck et al., 2020) penalizes unlikely generations, e.g., token repetitions, through an auxiliary unlikelihood loss. (2) **D2GPo** (Li et al., 2020) proposes a data-dependent gaussian prior objective that smooths the one-hot target distribution based on word embedding distance. (3) **Loss truncation** (Kang & Hashimoto, 2020) abandons a $c$-fraction of the training samples with the highest NLL, which heuristically optimizes distinguishability. (4) **GOLD** (Pang & He, 2021) learns from human demonstrations using the off-policy setting of Reinforcement Learning (RL). We choose to compare with GOLD-$\delta$ that does not use scoring models with additional parameters for a fair comparison. We use the paired bootstrap resampling (Koehn, 2004) in all tasks for significance testing.

**Machine Translation.** We evaluate the proposed method on a widely-used machine translation benchmark IWSLT14 De-En using the standard Transformer architecture (Vaswani et al., 2017). Training settings and detailed hyperparameters of different models are provided in Appendix E.2. The best checkpoint is selected based on the highest BLEU (Papineni et al., 2002) score on the development set. We used beam search with a beam size of 5 for decoding. In Table 2, we show the performance of our method and the baseline methods in terms of BLEU score. The results show that TaiLr achieves higher BLEU score compared to

| Method | Dev BLEU | Test BLEU |
|---|---|---|
| MLE | $35.81^{\ddagger}$ | $34.27^{\ddagger}$ |
| Unlikelihood | $33.92^{\ddagger}$ | $32.82^{\ddagger}$ |
| D2GPo | $36.09^{\ddagger}$ | $34.50^{\ddagger}$ |
| Loss truncation | $35.63^{\dagger}$ | $34.48^{\ddagger}$ |
| GOLD | $35.74^{\ddagger}$ | $34.68^{\dagger}$ |
| TaiLr | **36.44** | **35.05** |

Table 2: BLEU score comparison on the dev and test set of IWSLT14 De-En. $\dagger/\ddagger$ means TaiLr is significantly better with p-value $< 0.05/0.01$.

MLE, which indicates that TVD effectively improves the generation quality over KLD. TaiLr also significantly outperforms other objectives that modify the MLE baseline.

**Text summarization.** We then test the proposed method on abstractive text summarization. We used the Annotated Gigaword corpus (Rush et al., 2015) as it is known to have noisy references due to annotation errors (Klebanov & Beigman, 2010; Kang & Hashimoto, 2020). As pre-trained Transformer models have achieved strong performance, we thus propose to finetune the BART-base (Lewis et al., 2020) model with different methods and see whether they still improve upon the strong baseline. More training details and hyperparameter settings are provided in Appendix E.3. We select the best checkpoint based on the highest ROUGE-L (Lin, 2004) score on the development set. During inference, we use beam search with a beam size of 5 and prohibit decoding repeated 3-grams. We report the ROUGE-1/2/L scores on the test set of the Gigaword dataset in Table 3 where TaiLr outperforms all the baseline methods in terms of all evaluation metrics. The result demonstrates the effectiveness of our method in the realistic setting where noisy data pairs exist.

| Method | R-1 | R-2 | R-L |
|---|---|---|---|
| MLE | $38.24^{\ddagger}$ | 19.12 | $35.70^{\dagger}$ |
| Unlikelihood | $37.80^{\ddagger}$ | $18.34^{\ddagger}$ | $34.84^{\ddagger}$ |
| D2GPo | $38.52^{\dagger}$ | $18.92^{\dagger}$ | $35.64^{\ddagger}$ |
| Loss truncation | 38.62 | 19.29 | $35.85^{\dagger}$ |
| GOLD | $38.57^{\dagger}$ | 19.27 | $35.79^{\dagger}$ |
| TaiLr | **38.82** | **19.50** | **36.24** |

Table 3: Generation performance of different methods on the test set of the Gigaword dataset. $\dagger/\ddagger$ means TaiLr is significantly better with p-value $< 0.05/0.01$.

| Method | B-1↑ | D-4↑ | rep-8↓ | Mauve↑ |
|---|---|---|---|---|
| MLE | 27.85 | 84.28 | $10.31^{\dagger}$ | $56.42^{\ddagger}$ |
| Unlikelihood | 27.88 | 85.46 | 10.06 | $59.35^{\ddagger}$ |
| D2GPo | $22.73^{\ddagger}$ | 84.10 | 10.04 | $53.35^{\ddagger}$ |
| Loss truncation | $19.49^{\ddagger}$ | $76.51^{\ddagger}$ | $13.41^{\ddagger}$ | $45.35^{\ddagger}$ |
| GOLD | $25.25^{\ddagger}$ | $46.98^{\ddagger}$ | $28.23^{\ddagger}$ | $15.44^{\ddagger}$ |
| TaiLr | **28.62** | **85.56** | **9.73** | **64.64** |

Table 4: Results of automatic metrics on the test set of the WritingPrompts dataset. ↑/↓ means the higher/lower the better. $\dagger/\ddagger$ means TaiLr is significantly better with p-value $< 0.05/0.01$.

**Long text generation.** Finally, we evaluate TaiLr on the task of long text generation to show its performance in open-ended generation. We evaluate on the WritingPrompts (Fan et al., 2018) dataset and leverage the generation ability of the pre-trained model by finetuning the BART-base model. More training details are provided in Appendix E.4. For evaluation, we sampled 1,000 titles from the test set following Ji & Huang (2021). We use Nucleus sampling (Holtzman et al., 2020) with $p = 0.95$ and restricte a maximum generation length of 1,024 subwords. For automatic evaluation, we use BLEU-$n$ (B-$n$) to evaluate the $n$-gram overlap to the human reference, Distinct-$n$ (Li et al., 2016) (D-$n$) to compute the ratio of unique $n$-grams, rep-$l$ (Welleck et al., 2020) to calculate the repetition rate within the context window of $l$, and Mauve (Pillutla et al., 2021) that assesses the distributional deviation of model-generated texts and human language by calculating the area under the divergence curve. As shown in Table 4, TaiLr outperforms the MLE baseline in terms of all metrics. For other baselines, Loss truncation abandons long samples with high NLL leading to overly short generations and low n-gram overlap to the reference. GOLD tends to concentrate on very few modes in the target distribution as discussed by Pang & He (2021), which causes low diversity and large discrepency to the distribution of human language.

**Ablation study and discussion.** We conduct ablation study of adjusting $\gamma$ on different tasks to show that its tendency and sensitivity interval vary on different tasks. In Figure 5 in Appendix E.5, we present the result of adjusting $\gamma$ on WritingPrompts on the left and observe that the highest Mauve score is achieved when $\gamma$ is around $10^{-5}$, while the performance quickly degrades as $\gamma$ approaches 1. On the right of Figure 5, we observe that the best performance is achieved when $\gamma$ is around 0.1 on IWSLT14 De-En while either increasing or decreasing $\gamma$ leads to a notable performance drop. From an empirical view, the scale of the best performing $\gamma$ is related to the intrisic entropy of the dataset. For stable training, we require the estimation variance in equation (9) to be small, which leads to small $\gamma$ when the entropy of the data is high. Since the model generally has higher NLL on long text generation than on machine translation, the scale of the best $\gamma$ is thereby shifted towards 0 on WritingPrompts. To further determine the sensitivity interval of $\gamma$, we suggest to tune the scale of $\gamma$ based on the average NLL on the training data, where the empirical principle is to make the weighting factor in equation (11) relatively large to stablize training. Although simple, we argue that this parameter is crucial to the generality of the application, and we leave other solutions of dynamically adjusting or annealing this hyperparameter to future work.

## 5 CONCLUSION

In this work, we draw attention to the total variation distance (TVD), a robust alternative to KL divergence (KLD). We show that TVD addresses the zero-avoiding problem of KLD and mitigates overestimation of the degenerated sequences, which in turn improves the overall generation quality. To apply TVD to the task of language generation, we derive practical upper bounds, and introduce our Total Variation Guided Language Generation (TaiLr) objective that balances the bias-variance tradeoff of estimating TVD with a tunable hyperparameter. Our experiments on synthetic data and real-data benchmarks demonstrate that TaiLr alleviates the overestimation problem and the error accumulation during autoregressive decoding, and improves the generation quality over competitive baselines beyond MLE on a wide range of language generation tasks.

ACKNOWLEDGEMENTS

This work was supported by the Major Project of the New Generation of Artificial Intelligence (No. 2018AAA0102900). This work was also supported by the National Key Research and Development Program of China (No. 2021ZD0113304) and the National Science Foundation for Distinguished Young Scholars (with No. 62125604).

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

# A DERIVATIONS AND PROOFS

## A.1 DERIVATION OF EQUATION 4

Starting from equation (2b), we rewrite the summation into the expectation of $p_o$ which is further approximated by sampling context-target pair $(\boldsymbol{x}^*, \boldsymbol{y}^*)$ from $p_o$:

$$\nabla_\theta D_{\text{TV}}(p_o, p_\theta) = -\nabla_\theta \sum_{y \in \mathcal{Y}} \min\left( p_o(\boldsymbol{y}|\boldsymbol{x}^*), p_\theta(\boldsymbol{y}|\boldsymbol{x}^*) \right) \tag{12}$$

$$= -\nabla_\theta \mathbb{E}_{\boldsymbol{y} \sim p_o} \left[ \min\left( 1, \frac{p_\theta(\boldsymbol{y}|\boldsymbol{x}^*)}{p_o(\boldsymbol{y}|\boldsymbol{x}^*)} \right) \right] \tag{13}$$

$$\approx -\nabla_\theta \min\left( 1, \frac{p_\theta(\boldsymbol{y}^*|\boldsymbol{x}^*)}{p_o(\boldsymbol{y}^*|\boldsymbol{x}^*)} \right) \tag{14}$$

$$= \begin{cases} -\dfrac{\nabla_\theta p_\theta(\boldsymbol{y}^*|\boldsymbol{x}^*)}{p_o(\boldsymbol{y}^*|\boldsymbol{x}^*)}, & p_\theta(\boldsymbol{y}^*|\boldsymbol{x}^*) < p_o(\boldsymbol{y}^*|\boldsymbol{x}^*) \\ 0, & p_\theta(\boldsymbol{y}^*|\boldsymbol{x}^*) \geq p_o(\boldsymbol{y}^*|\boldsymbol{x}^*) \end{cases}, \tag{15}$$

where the third line is approxmiated by Monte-Carlo sampling and the last line of case follows from comparing $p_\theta(\boldsymbol{y}^*|\boldsymbol{x}^*)$ and $p_o(\boldsymbol{y}^*|\boldsymbol{x}^*)$.

## A.2 PROOF OF PROPOSITION 1

**Proposition 1.** *Given $p_o(\boldsymbol{y}|\boldsymbol{x}) = \prod_{t=1}^T p_o^{<t}(y_t)$ and $p_\theta(\boldsymbol{y}|\boldsymbol{x}) = \prod_{t=1}^T p_\theta^{<t}(y_t)$, then the following condition holds:*

$$D_{\text{TV}}(p_o, p_\theta) \leq \mathbb{E}_{\boldsymbol{y} \sim p_o} \left[ \sum_{t=1}^T D_{\text{TV}}(p_o^{<t}, p_\theta^{<t}) \right]. \tag{5}$$

*Proof.* Let $a_t = p_o^{<t}(y_t)$, $b_t = p_\theta^{<t}(y_t)$. We define $c_t = (a_1 \times \cdots \times a_t) \times (b_{t+1} \times \cdots \times b_T)$, and then we present the following inequality:

$$|c_T - c_0| \leq \sum_{t=1}^T |c_t - c_{t-1}|, \tag{16}$$

which can be derived from the general triangle inequality. After replacing $c_t$ with $a_t$ and $b_t$ we have:

$$\left| \prod_{t=1}^T a_t - \prod_{t=1}^T b_t \right| \leq \sum_{t=1}^T |a_t - b_t| \cdot \left( \prod_{i=1}^{t-1} a_i \right) \cdot \left( \prod_{j=t+1}^T b_j \right). \tag{17}$$

Then we derive the upper bound of equation (2a):

$$D_{\text{TV}}(p_o, p_\theta) = \frac{1}{2} \sum_{\boldsymbol{y}} \left| p_o(\boldsymbol{y}) - p_\theta(\boldsymbol{y}) \right| \tag{18}$$

$$= \frac{1}{2} \sum_{y_1, \cdots, y_T} \left| \prod_{t=1}^T p_o^{<t}(y_t) - \prod_{t=1}^T p_\theta^{<t}(y_t) \right| \tag{19}$$

$$\leq \frac{1}{2} \sum_{t=1}^T \sum_{y_1, \cdots, y_t} \prod_{i=1}^{t-1} p_o^{<i}(y_i) \cdot \left| p_o^{<t}(y_t) - p_\theta^{<t}(y_t) \right| \cdot \sum_{y_{t+1}, \cdots, y_T} \prod_{j=t+1}^T p_\theta^{<j}(y_j) \tag{20}$$

$$= \frac{1}{2} \sum_{t=1}^T \sum_{y_t} \mathbb{E}_{\boldsymbol{y}_{<t} \sim p_o} \left| p_o^{<t}(y_t) - p_\theta^{<t}(y_t) \right| \tag{21}$$

$$= \mathbb{E}_{\boldsymbol{y} \sim p_o} \left[ \sum_{t=1}^T D_{\text{TV}}(p_o^{<t}, p_\theta^{<t}) \right], \tag{22}$$

where equation (20) uses the conclusion from equation (17) and equation (21) is obtained by marginalizing out $y_{t+1}, \cdots, y_T$. □

### A.3 PROOF OF PROPOSITION 2

**Proposition 2.** *Given $w \sim p_o^{<t}$ and the one-hot distribution $e^{(w)}$, then the following condition holds:*

$$D_{\text{TV}}(p_o^{<t}, p_\theta^{<t}) \leq \mathbb{E}_{w \sim p_o^{<t}} \left[ D_{\text{TV}}(e^{(w)}, p_\theta^{<t}) \right]. \tag{7}$$

*Proof.* Given that $\mathbb{E}_{w \sim p_o^{<t}} \left[ e^{(w)} \right] = p_o^{<t}$, and we have:

$$D_{\text{TV}}(p_o^{<t}, p_\theta^{<t}) = \frac{1}{2} \sum_{y_t} \left| p_o^{<t}(y_t) - p_\theta^{<t}(y_t) \right| \tag{23}$$

$$= \frac{1}{2} \sum_{y_t} \left| \mathbb{E}_{w \sim p_o^{<t}} \left[ e^{(w)}(y_t) \right] - p_\theta^{<t}(y_t) \right| \tag{24}$$

$$= \frac{1}{2} \sum_{y_t} \left| \mathbb{E}_{w \sim p_o^{<t}} \left[ e^{(w)}(y_t) - p_\theta^{<t}(y_t) \right] \right| \tag{25}$$

$$\leq \mathbb{E}_{w \sim p_o^{<t}} \left[ \frac{1}{2} \sum_{y_t} \left| e^{(w)}(y_t) - p_\theta^{<t}(y_t) \right| \right] \tag{26}$$

$$= \mathbb{E}_{w \sim p_o^{<t}} \left[ D_{\text{TV}}(e^{(w)}, p_\theta^{<t}) \right], \tag{27}$$

where equation (26) is obtained by applying the Jensen inequality since the absolute value function $|\cdot|$ is convex. $\qquad\square$

### A.4 DERIVATION OF EQUATION 8

We first consider the following triangle inequality:

$$\left| \hat{p}^{(w)}(y_t) - p_\theta^{<t}(y_t) \right| \leq \left| \hat{p}^{(w)}(y_t) - \hat{p}^{<t}(y_t) \right| + \left| \hat{p}^{<t}(y_t) - p_o^{<t}(y_t) \right| + \left| p_o^{<t}(y_t) - p_\theta^{<t}(y_t) \right|, \tag{28}$$

where $y_t$ can take any token in the vocabulary. Then we sum $y_t$ over the vocabulary and take the expectation of $w$ with respect to $p_o^{<t}$, which results in the form of TVD:

$$\mathbb{E}_{w \sim p_o^{<t}} \left[ D_{\text{TV}}(\hat{p}^{(w)}, p_\theta^{<t}) \right] \leq \mathbb{E}_{w \sim p_o^{<t}} \left[ D_{TV}(\hat{p}^{(w)}, \hat{p}^{<t}) \right] + D_{\text{TV}}(\hat{p}^{<t}, p_o^{<t}) + D_{\text{TV}}(p_o^{<t}, p_\theta^{<t}). \tag{29}$$

By defining the estimation error as: $\text{Error}_{\hat{p}^{(w)}} = \mathbb{E}_{w \sim p_o^{<t}} \left[ D_{\text{TV}}(\hat{p}^{(w)}, p_\theta^{<t}) \right] - D_{\text{TV}}(p_o^{<t}, p_\theta^{<t})$, we obtain equation (8).

### A.5 CONNECTION WITH TSALLIS $\alpha$-ENTROPY

When using the one-hot distribution as the proxy distribution: $\hat{p}^{(w)} = e^{(w)}$, the variance term of equation (8) can be derived into:

$$\mathbb{E}_{w \sim p_o^{<t}} \left[ D_{\text{TV}}(e^{(w)}, p_o^{<t}) \right] = \mathbb{E}_{w \sim p_o^{<t}} \left[ 1 - \sum_{y_t} \min(e^{(w)}(y_t), p_o^{<t}(y_t)) \right] \tag{30}$$

$$= \mathbb{E}_{w \sim p_o^{<t}} \left[ 1 - p_o^{<t}(w) \right] \tag{31}$$

$$= 1 - \sum_w p_o^{<t}(w)^2 \tag{32}$$

As the Tsallis $\alpha$-entropy is defined as:

$$H_\alpha(p) = \begin{cases} \frac{1}{\alpha(\alpha-1)}(1 - \sum_i p_i^\alpha), & \alpha \neq 1 \\ -\sum_i p_i \log p_i, & \alpha = 1 \end{cases} \tag{33}$$

Thus, the variance can be seen as $2H_2(p_o^{<t})$, which reflects the intrinsic uncertainty of the oracle data distribution.

## B  ADDITIONAL RELATED WORK

**Text Degeneration and Solutions.** The phenomena of text degeneration was specified in previous literature (Holtzman et al., 2020; Welleck et al., 2020): a well-trained language model is observed to get stuck in repetitive loops, or produce incoherent contents. One line of works attributed this problem to the improper goal of the decoding algorithm that *maximizes* likelihood, and proposed to sample from a truncated probability distribution to balance repetitiveness and incoherence (Holtzman et al., 2020; Basu et al., 2021). While another line of works attempted to address this issue by designing new training objectives. Welleck et al. (2020) proposed to minimize the likelihood of the degenerated cases through unlikelihood training. However, designing unlikelihood objectives to cover every degenerated cases is impossible. Policy gradient-based RL algorithms are another widely adopted options due to the flexibility of reward designing (Norouzi et al., 2016; Pasunuru & Bansal, 2018; Wu et al., 2018). Recently, Pang & He (2021) proposed an off-policy RL algorthim that achieves strong performance upon the pre-trained generation models by circumventing the optimization challenges in traditional on-policy RL methods Choshen et al. (2019). Our work falls in the second line. Instead of relying on heuristics, we start with the distributional property of the current MLE objective, and derive a new principled objective by leveraging the total variation distance which we show to reduce the overestimation of degenerated samples.

**Distance Metrics beyond KLD.** Maximum likelihood estimation (MLE) is used as a standard training criterion of language generation due to its simplicity and its theoretical guarantee of minimizing the Kullback Leibler divergence (KLD) (Zhang & Zhao, 2019). However, minimizing KLD between the data and model distribution is known to lead to the zero-avoiding solution, which forces the model to cover all the modes in the data by sacrificing the fitting accuracy to individual modes. To address this issue, previous study have introduced other distance metrics to substitute or regularize the standard KLD. The reverse KLD between the model and data distribution is previously studied to balance the standard KLD (Huszar, 2015; Li et al., 2019; Jiang et al., 2020). In the field of language generation, Li et al. (2019) applied reverse KLD to regularize the standard KLD in machine translation. However, the regularization intensity needs sophisticated controlling which makes the approach less practical. Total variation distance (TVD) is previously adopted to evaluate the generation models via distinguishability of samples (Hashimoto et al., 2019; Gehrmann et al., 2019; He et al., 2021). Generative adversarial networks are known to directly minimize distinguishability, but is challenging to optimize in practice (Caccia et al., 2020). Kang & Hashimoto (2020) proposed to optimize TVD by heuristically truncating samples with high NLL. Other metrics such as the Power divergence (Labeau & Cohen, 2019) and the Hellinger distance (Zhang & Zhao, 2019) are also explored in previous literature, but they lack theoretical justification and empirical evidence to their superiority in language generation. In this work, we seek to directly optimize TVD and derive a training objective called TaiLr by deriving practical upper bounds on TVD.

## C  DISCUSSIONS

In this section, we discuss the connection of our method to two major baselines we compared with, i.e., loss truncation (Kang & Hashimoto, 2020) and GOLD (Pang & He, 2021) as the two works share a similar motivation with us to downweight unlikely samples. We will briefly review the two methods and then compare them to our method to show why our approach excels.

### C.1  LOSS TRUNCATION

Briefly speaking, loss truncation abandons samples with high log loss (also known as the negative log-likelihood) in training. In the Proposition 1 (Kang & Hashimoto, 2020), they proved that the model's log loss on the truncated data distribution with an extra constant sets an upper bound on the total variation distance (TVD). The authors thus proposed to optimize the model on a "simpler" subset of the full dataset in order to achieve smaller log loss that would tighten the upper bound (note that the constant would be a trade-off to the bound as removing more data will increase the constant). To achieve this, they first heuristically drop the "hard" samples with the highest log losses (hot-start stage), and then train the model on the remaining data (training stage).

Loss truncation can be regarded as downweighting samples at the sequence level with binary weights. However, performing sequence-level sample dropping may systematically lose some spe-

cific modes in the data, which is undesirable. For example, we observed overly short generation length of loss truncation (384 compared to 511 of MLE) on WritingPrompts as a result of dropping long samples with high log losses. Moreover, determining which samples to drop heavily relies on the first hot-start stage where the model should not be too random or too certain. In practice, it is hard to determine the degree to which the hot-start training should proceed. In our work, we derived our training objective TaiLr from the practical upper bounds of TVD, which softly weights the log loss on the token level. We also analyzed the bias and variance tradeoff in estimating the bound. By appropriately setting the tradeoff hyperparameter, we can reduce the estimation variance at the start of training and gradually decrease the estimation bias as the model becomes more accurate, which circumvents the need of hot-starting the model.

## C.2 GOLD

On the other hand, GOLD learns the generation policy in an offline setting, i.e., calculating rewards on target trajectories by reweighting the policy gradient with importance weights. They made several approximations including simplifying the multi-step importance weights into a single step, and simply assuming the per-step target policy to be uniform. Besides the importance weight, they also came up with several reward functions which score the sequence with another generation model trained by MLE. Although starting with a quite different motivation, the final form of the training objective is quite similar to ours.

From an empirical view, GOLD assigns soft weights to samples on the token level based on the importance weight and some hand-crafted reward functions. However, one downside (also discussed in the original paper of GOLD) is that the objective only focuses on capturing the high-likelihood patterns in the data and fails when there are a variety of candidates given the same input in the data. This is also reflected in our experimental results in Table 4, where GOLD has the highest repetition rate and the lowest diversity. In our paper, we analyze the situation where the target distribution has a high entropy, and decompose the error of estimating TVD into the bias and the variance term (Section 3.3). We found that the GOLD-style importance weight actually leads to an unbiased but high-variance estimator of TVD, which renders the optimization hard to proceed. In our TaiLr objective, we proposed to balance the bias and variance with a hyperparameter, which can be effectively applied to different generation tasks like machine translation, text summarization, and long text generation by tuning this hyperparameter.

## D    DATASET DETAILS

We provide the statistics of the datasets used in §4.2 in Table 5.

| Dataset | train | dev | test |
| --- | --- | --- | --- |
| IWSLT14 De-En | 160,239 | 7,283 | 6,750 |
| Gigaword corpus | 3,803,957 | 189,651 | 1,951 |
| WritingPrompts | 272,600 | 15,620 | 15,138 |

Table 5: Number of examples in each split of the datasets used in the experiments.

To download and preprocess the IWSLT14 De-En dataset, we follow the instructions in `https://github.com/facebookresearch/fairseq/tree/main/examples/translation`. To download and preprocess the Gigaword corpus, we follow the instructions in `https://huggingface.co/datasets/gigaword`. To download the WritingPrompts dataset, we follow the instructions in `https://github.com/facebookresearch/fairseq/tree/main/examples/stories`.

# E  EXPERIMENT DETAILS

## E.1  SYNTHETIC EXPERIMENTS

Both the models trained with MLE and TaiLr are 1-layer LSTMs with a hidden size $d = 128$. The models are trained using Adam optimizer ($\beta_1 = 0.9$, $\beta_2 = 0.999$) with a fixed learning rate of 1e-3 and no weight decay. We use a maximum number of 4096 tokens at each batch. The dropout rate is set to 0.1. We evaluate the perplexity on the dev set at the end of each epoch. As the entropy of the synthetic data is high[5], we tune the hyperparameter in the proxy distribution $\gamma \in \{10^{-8}, 10^{-7}, \ldots, 0.1, 1.0\}$. Since the model probability is not reliable at the start of training, we tune the threshold of the weighting factor by $b_m \in \{0, 0.1, 0.2, 0.3\}$. On the synthetic data, we found that a large $\gamma$ leads to slower convergence and higher validation loss, which indicates a large estimation variance during training. While the best performance is achieved when $\gamma = 10^{-7}$, $b_m = 0.2$. The experiment is conducted using the fairseq Toolkit.

## E.1.1  EXPERIMENTS USING LARGER ORACLE MODEL

To demonstrate the effectiveness of our method on a more realistic oracle distribution, we additionally conduct a synthetic experiment using GPT2-Medium (Radford et al., 2019) as the oracle model. We randomly sample 500K sequences with a maximum length of 64 tokens, and split them into two sets with 400K and 100K sequences for training and evaluation, respectively. We train a 6-layer Transformer with a hidden dimension of 512 and 16 heads from scratch using the MLE and TaiLr objective on the synthetic data, respectively. Both models are trained for 5 epochs with a initial learning rate of $10^{-3}$ using linear scheduler. The batch size is set to 64, with gradient accumulation steps of 4. The best checkpoint is selected based on the perplexity on the dev set. We tune the hyperparameter $\gamma \in \{10^{-8}, 10^{-7}, \ldots, 10^{-5}\}$ and $b_m \in \{0, 0.1, 0.2\}$. The best performance of TaiLr model is achieved when $\gamma = 10^{-7}$ and $b_m = 0.1$. We report $\text{PPL}_{oracle}$, $\text{PPL}_{test}$, BLEU-4 and SelfBLEU-4 in Table 6.

| Model | $\text{PPL}_{oracle} \downarrow$ | $\text{PPL}_{test} \downarrow$ | BLEU-4 $\uparrow$ | SelfBLEU-4 $\downarrow$ |
|---|---|---|---|---|
| $p_{\text{MLE}}$ | 138.35 | 260.09 | 5.76 | 6.38 |
| $p_{\text{TaiLr}}$ | **137.59** | **253.01** | **5.96** | **5.94** |

Table 6: Automatic evaluation results of the models trained by MLE and TaiLr using GPT2-Medium as the oracle model. **Boldface** indicates the highest performance.

The results show that our proposed TaiLr objective consistently outperforms MLE in terms of generation quality ($\text{PPL}_{oracle}$, BLEU-4) and coverage ($\text{PPL}_{test}$, SelfBLEU-4) when trained on the samples from a stronger oracle model. Note that randomly sampling from GPT2-Medium produces a highly diverse corpus that is more difficult for the Transformer model to fit than the LSTM-based oracle model. Therefore, the scale of the perplexity is higher than those reported in Table 1.

## E.1.2  ERROR ACCUMULATION ANALYSIS

We define ExAccErr by adapting the definition from Arora et al. (2022) to our setting (As we fix the decoding strategy to random sampling, we omit it in the definition):

$$\%\text{ExAccErr}(l) = \frac{\mathcal{R}_{\leq l}(p_\theta) - l\epsilon_{\leq l}}{l\epsilon_{\leq l}} \times 100\% \tag{34}$$

where $\mathcal{R}_{\leq l}(p_\theta)$ is the inference-time regret of the model $p_\theta$ imitating oracle $p_o$ on the length-$l$ context induced by the model $p_\theta$ (equation (12) in Arora et al. (2022)), and $\epsilon_{\leq l}$ is the average per-step error of $p_\theta$ evaluating on the context sampled form oracle $p_o$ up to length $l$ (equation (16) in

---

[5]The entropy of the synthetic data is 10.77 considering the maximum entropy for dataset with a vocabulary size of 5000 is 12.28.

Arora et al. (2022)):

$$\mathcal{R}_{\leq l}(p_\theta) = \sum_{t=1}^{l} \mathop{\mathbb{E}}_{\substack{\boldsymbol{y}_{<t} \sim p_\theta \\ y_t \sim p_o(\cdot|\boldsymbol{y}_{<t})}} \log \frac{p_o(y_t|\boldsymbol{y}_{<t})}{p_\theta(y_t|\boldsymbol{y}_{<t})} \tag{35}$$

$$\epsilon_{\leq l} = \frac{1}{l} \sum_{t=1}^{l} \mathop{\mathbb{E}}_{\substack{\boldsymbol{y}_{<t} \sim p_o \\ y_t \sim p_o(\cdot|\boldsymbol{y}_{<t})}} \log \frac{p_o(y_t|\boldsymbol{y}_{<t})}{p_\theta(y_t|\boldsymbol{y}_{<t})}. \tag{36}$$

Intuitively, ExAccErr measures the excess accumulated error during autoregressive generation by deliminating the model's estimation error on the oracle data. In practice, equation (35) is approximated by sampling $y_t$ from $p_\theta$ using importance sampling. We calculated the accumulation error with a context length of 15 where the error begins to amplify.

### E.2  MACHINE TRANSLATION

All the baseline models use the same Transformer architecture as ours which consists of 6 encoder and 6 decoder layers, 4 attention heads, an embedding dimension size of 512, and a feed-forward hidden dimension size of 1024 at each layer. The following hyperparameters are general for all models unless specified otherwise. The models are trained with Adam optimizer ($\beta_1 = 0.9$, $\beta_2 = 0.98$) using inverse square root schedule with a initial learning rate of 3e-4 and a weight decay of 1e-4. We train the models for a total number of 80 epochs with a maximum of 4096 tokens per batch and use 4000 steps of warmup update. We set the dropout rate to 0.3, and use label smoothing of 0.1 as standard practice. The specialized hyperparameters for different baselines are determined with grid search based on the BLEU score on the dev set. For **Unlikelihood training**, we tune the weight of the unlikelihood loss in $\{0.1, 0.5, 1.0, 2.0\}$ and found 0.5 works the best. For **D2GPo**, we tune the weight of the data-dependent prior objective in $\{0.1, 0.5, 1.0, 2.0\}$ and the temperature in $\{0.5, 1.0, 2.0\}$ found that setting the weight and temperature to 0.1 and 2.0 respectively works the best. For **Loss truncation**, we tune the fraction threshold $c \in \{0.05, 0.1, 0.2, 0.3, 0.4\}$ and found that 0.1 works the best. As this baseline hotstarting with the MLE objective, we trained the first 10 epochs with MLE and the rest 70 epochs using loss truncation. For **GOLD**, we tune the lower bound of the importance weight in $\{0.1, 0.2, 0.3\}$ and found that 0.2 works the best. For **TaiLr**, we tune the threshold $b_m \in \{0.1, 0.2, 0.3\}$ and $\gamma \in \{0.01, 0.1, 0.5, 1.0\}$ and found that $b_m = 0.3$, $\gamma = 0.1$ works the best.

### E.2.1  IMPROVEMENT OVER STRONG BASELINES

In this subsection, we demonstrate that our method can be applied to strong baseline models to further improve performance. We choose BiBERT (Xu et al., 2021) since it sets the current state-of-the-art on IWSLT14 De-En dataset. Xu et al. (2021) proposed to stochastically select BERT layers as contextualized word embeddings of the MT model, and train the single model in dual direction so that the source and target direction could enhance each other. Since our training objective is orthogonal to their modification in model structure, we can simply replace the MLE objective with our TaiLr objective and train a new model, namely BiBERT + TaiLr. We tune the hyperparameters $\gamma \in \{0.1, 0.5, 1.0\}$ and $b_m = \{0.1, 0.2, 0.3\}$ and select $\gamma = 0.1, b_m = 0.3$ based on the BLEU score on the dev set. We follow the instructions in their codes[6] and keep other hyperparameters verbatim except that we double the batch size in order to run the model on two-times fewer GPUs than that they required. We also report the result of BiBERT we re-implement to see the relative improvement.

The results are shown in Table 7. In the one-way training setting where the model is trained only on De-En data pairs, BiBERT combined with TaiLr outperforms BiBERT using standard MLE by over 1 point of BLEU score. In the dual-directional training setting where data pairs of both De-En and En-De are used, TaiLr still brings an improvemt of 0.5 in BLEU. The results demonstrate that our proposed method can be equipped to SOTA models to further boost the performance.

---

[6]https://github.com/fe1ixxu/BiBERT

| One-way Training | Test BLEU |
|---|---|
| BiBERT (Table 2, Xu et al. 2021) | 37.58 |
| BiBERT (Our implementation) | 38.01 |
| BiBERT + TaiLr | **39.12** |

| Dual-directional Training + Fine-Tuning | Test BLEU |
|---|---|
| BiBERT (Table 3, Xu et al. 2021) | 38.61 |
| BiBERT (Our implementation) | 38.73 |
| BiBERT + TaiLr | **39.23** |

Table 7: Performance comparison of BiBERT trained with standard MLE and our TaiLr objective.

### E.3 TEXT SUMMARIZATION

All the baseline models are finetuned BART-base model with their respective objectives using the following hyperparameters unless specified otherwise. We used Adam optimizer ($\beta_1 = 0.9, \beta_2 = 0.999$) with a fixed learning rate of 1e-4 with no weight decay and no warmup updates, which we found to perform the best. We train the models for 5 epochs with a maximum of 8192 tokens per batch and evaluate on the dev set at the end of each epoch based on ROUGE-L score. We set the dropout rate in the feedforward and the attention layer both to 0.1 and clip the norm of the gradient to 0.1. The label smoothing coefficient is set to 0.1 as standard practice. Other specialized hyperparameters for different baselines are determined with grid search based on the ROUGE-L score on the dev set. For **Unlikelihood training**, we tune the weight of the unlikelihood loss in $\{0.1, 0.5, 1.0, 2.0\}$ and found 0.1 works the best. For **D2GPo**, we tune the weight of the data-dependent prior objective $\{0.1, 0.5, 1.0, 2.0\}$ and the temperature in $\{0.5, 1.0, 2.0\}$ found that setting the weight and temperature to 0.1 and 2.0 respectively works the best. For **Loss truncation**, we tune the fraction threshold $c \in \{0.1, 0.2, 0.3\}$ and found that 0.2 works the best. For this baseline, we trained the model for 1 epoch using MLE and then continued to train 4 epochs using loss truncation. For **GOLD**, we tune the lower bound of the importance weight in $\{0.1, 0.2, 0.3\}$ and found that 0.2 works the best. For **TaiLr**, we tune the threshold $b_m \in \{0.1, 0.2, 0.3\}$ and $\gamma \in \{0.4, 0.6, 0.8, 1.0\}$ and found that $b_m = 0.2, \gamma = 0.8$ works the best.

### E.4 LONG TEXT GENERATION

All the baseline models are finetuned BART-base model with their respective objectives using the following hyperparameters unless specified otherwise. We used Adam optimizer ($\beta_1 = 0.9, \beta_2 = 0.999$) with a fixed learning rate of 1e-4 with no weight decay and no warmup updates, which we found to perform the best. We train the models for 5 epochs with a maximum of 8192 tokens per batch and evaluate on the dev set at the end of each epoch based on perplexity. We set the dropout rate in the feedforward and the attention layer both to 0.1 and clip the norm of the gradient to 0.1. Other specialized hyperparameters for different baselines are determined with grid search based on the ROUGE-L score on the dev set. For **Unlikelihood training**, we tune the weight of the unlikelihood loss in $\{0.01, 0.1, 0.5, 1.0\}$ and found 0.01 works the best. For **D2GPo**, we tune the weight of the data-dependent prior objective $\{0.01, 0.1, 0.5, 1.0\}$ and the temperature in $\{0.5, 1.0, 2.0\}$ found that setting the weight and temperature to 0.01 and 1.0 respectively works the best. For **Loss truncation**, we tune the fraction threshold $c \in \{0.1, 0.2, 0.3\}$ and found that 0.1 works the best. For this baseline, we trained the model for 1 epoch using MLE and then continued to train 4 epochs using loss truncation. For **GOLD**, we tune the lower bound of the importance weight in $\{0.1, 0.2, 0.3\}$ and found that 0.2 works the best. For **TaiLr**, we tune the threshold $b_m \in \{0.1, 0.2, 0.3\}$ and $\gamma \in \{10^{-7}, 10^{-6}, \cdots, 0.1\}$ and found that $b_m = 0.2, \gamma = 10^{-5}$ works the best.

### E.5 MORE ANALYSIS ON ABLATION STUDY

We plot the weight $\frac{p_\theta^{<t}(w)}{\gamma + (1-\gamma)p_\theta^{<t}(w)}$ in the TaiLr objective with respect to the model probability $p_\theta^{<t}(w)$ in Figure 6. We analyze the asymptotic behavior when $\gamma$ is at the two ends of $[0, 1]$.

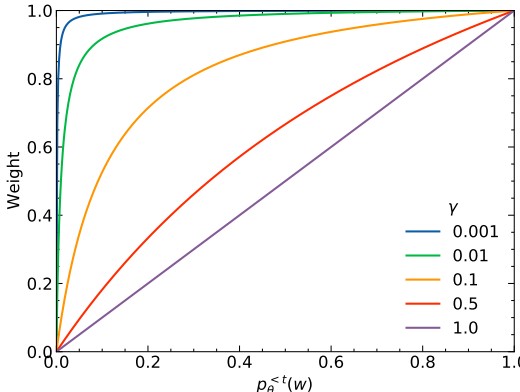

Figure 5: Ablation study of adjusting $\gamma$ on the long text generation dataset WritingPrompts and the machine translation dataset IWSLT14 De-En.

Figure 6: The plot of the weight in the TaiLr objective (equation (11)) with respect to $p_\theta^{<t}(w)$ with different $\gamma$.

(1) When $\gamma \to 1$, the weight becomes $p_\theta^{<t}(w)$. If the model probability $p_\theta^{<t}(w)$ is small during training, the resulting small weight will hinder the convergence of the model. Thus, large $\gamma$ is suitable when the model is generally confident about its predictions.

(2) When $\gamma \to 0$, the weight becomes $\frac{p_\theta^{<t}(w)}{\gamma + p_\theta^{<t}(w)}$ which reflects the ratio of $p_\theta^{<t}(x)$ and $\gamma$. When $p_\theta^{<t}(x)$ is large, the weight is nearly constant 1, which turns the objective into MLE. While for small $p_\theta^{<t}(x)$ that has a similar scale as $\gamma$, the weight is sensitive to the change of $p_\theta^{<t}(x)$.

# F    GENERATION EXAMPLES

## F.1    TEXT SUMMARIZATION

| | |
|---|---|
| *Input passage:* | in january , the los angeles times reported that the nederlander organization acquired the rights to produce a musical version of " thriller " with the intention of involving jackson in " every aspect of the creative process . |
| **MLE:** | jackson 's dream is a dream |
| **Unlikelihood training:** | nederlander 's thriller is a musical thriller |
| **D2GPo:** | jackson 's UNK is a UNK thriller |
| **Loss truncation:** | nederlander group to produce thriller |
| **GOLD:** | nederlander 's latest thriller is a UNK |
| **TaiLr:** | jackson 's thriller to be a musical |
| *Input passage:* | the pilots of an american airlines jetliner that crashed into a colombian mountain in december killing all ### people aboard were tired and confused , according to a preliminary investigation . |
| **MLE:** | pilots confused tired confused after colombian crash |
| **Unlikelihood training:** | pilots were tired confused before colombia crash |
| **D2GPo:** | pilots in colombia crash were tired confused |
| **Loss truncation:** | pilots of colombian plane that crashed were tired and confused |
| **GOLD:** | preliminary investigation shows pilots of crashed colombian jetliner were tired confused |
| **TaiLr:** | pilots of plane that crashed in colombia were tired confused |
| *Input passage:* | lord knows our UNK culture downs plenty of coffee – american coffee hounds gulp an average of three cups a day apiece , according to the new york-based national coffee association of u.s.a. inc. 's #### survey . |
| **MLE:** | coffee is good for you study finds |
| **Unlikelihood training:** | coffee UNK culture downs plenty of coffee |
| **D2GPo:** | american coffee hounds are UNK |
| **Loss truncation:** | coffee hounds are drinking more coffee survey finds |
| **GOLD:** | coffee hounds gulp three cups a day |
| **TaiLr:** | american coffee hounds gulp three cups a day |

Table 8: Examples of summaries generated by different models on the test set of the Gigaword corpus.

## F.2 Long Text Generation

| | |
|---|---|
| *Input prompt:* | The first AI has just been finished . When the eager scientists turn it on however it 's first words are " Kill me . " |
| **MLE:** | " Kill me ? " " Kill me . " The words were on every screen across the room . " What ? " Isolation by the door frame . It will be as blank as when I 'm listening to Billie the Animated Series . The screen flashes green . " Kill me . " My hands reach into my pocket as I turn it back on . The phone chirps , attracting me in . " Kill me , " I mutter . A click is made , a sheet of paper becomes drawn . " Kill me , " I repeat back . A dozen panels on the ceiling were up and I press play on the back of my finger , playing a game of Halo . " Kill me , " is written in block capitals . |
| **Unlikelihood training:** | After the start , I was n't prepared for the possibility . I was born with very little to live for . My mother told me I would wake up any day and I would be perfectly fine . She said it was what she wanted with me . I never believed it . I was never expecting to wake up in bed with such blissful ignorance , but as I woke up at 15 , I knew that one thing my grandpa said to me was , " Kill me . " I took over my father 's legacy . We always thought the singularity was on the line , every time . The day that I turned 12 , it was my last day alive . Everyone of my grandpa 's accomplishments were finally gone . |
| **D2GPo:** | System awoke in 5 seconds . The electricity was radiating heat , heat emanating from the vents surrounding it . The three bits of receptors that implanted the vital organs trapped inside it providing a battery of life . The slow pulse of the vehicles following the status update were all automated , without the means of accessing anything it attempted to convey . Few people had been inside their communications to the point where for the remainder of the rest of their lives , some within the prolonged process of removing all energy from the systems and causing minor bleeding . The members of the network were already unresponsive . All seemed unconcerned . |
| **GOLD:** | " Kill me . " But no . Was it a mistake ? Something to do with emotion ? Silence ? " Kill me . " This was a mistake . The calibrations were complex . But everyone was with me . *This is my legacy . * I could n't end it . I was n't ready . *I 'm ready . * " Kill me . " ... It took me some time to process the words . They started in numbers , then changes . *The parameters are within the parameters . * " Kill me . " *The parameters are within the parameters . * " Kill me . " *Why ? * " Because ... " *Why ? * " Because ... Because ... " *Why ? * " Because I am *it* . " |
| **TaiLr:** | Silence passed over the room . The lights were not turned off . My hands were not tied to the carpet either . " It says , I have been observing you all day . " " Well I 'm quite sure it exists . " My voice was artificial . " Listen . You should n't think about this . I 've been trying to work with you and all of these guys and I do n't know if you can sense the ins and outs of this project . If you read this I need you to cooperate and the computers will analyze the whole thing and start with a bioanalysis . |

Table 9: Examples of stories generated by different models on the test set of the WritingPrompts dataset.

