# OpenReview forum: "Tailoring Language Generation Models under Total Variation Distance"
_ICLR.cc/2023/Conference — ICLR 2023 notable top 5%_

### Official Review · Reviewer_Zays · 2022-10-25

**Confidence:** 4
**Clarity, Quality, Novelty And Reproducibility:** The paper is clearly written, high qu…
**Correctness:** 4
**Technical Novelty And Significance:** 3
**Empirical Novelty And Significance:** 3
**Recommendation:** 8

**Strength And Weaknesses:**

Strengths:
-Novel method
-Thorough empirical results (both real and synthetic)
-Well written

**Summary Of The Paper:**

The paper studies training language models using total variation distance as opposed to maximum likelihood. While the total variation distance idea was previously proposed by Kang and Hashimoto 2020, they optimized a different upper bound via loss truncation and Pinsker's Inequality.

In this work, the authors propose an alternative upper bound that is novel and interesting by tackling specific challenges with optimizing the TVD loss (i.e. lack of token level factorization, and explicit demand of data probability distribution)

The authors provide empirical results both on real and synthetic data. In particular in machine translation (IWSLT De-En), text summarization (GigaWord) and open ended generation (Writing Prompts), the authors' approach outperforms not only MLE but several other techniques in the literature (unlikelihood, D2GPo, loss truncation, and GOLD).


**Summary Of The Review:**

I support accepting the paper. It is quite interesting, novel and tackles a very important problem in language models.

---

> ### Author Response · Authors · 2022-11-16
> **Response to Reviewer Zays**
>
> We thank the reviewer for the comment! We will continue to improve our work and release our code for future work!

---

### Official Review · Reviewer_PwuP · 2022-10-25

**Confidence:** 3
**Correctness:** 3
**Technical Novelty And Significance:** 3
**Empirical Novelty And Significance:** 2
**Recommendation:** 8

**Clarity, Quality, Novelty And Reproducibility:**

The code is attached but I don't see README. The code is simply a edited clone of fairseq which contains a large number of files.

**Strength And Weaknesses:**

Experiments are on a synthetic task as well as regular tasks including machine translation and summarization. Experiments are also done on long text generation as well, and I’m glad to see that it works well.


I’m glad that the paper involves discussion of Kang and Hashimoto (2020) as well as Pang and He (2021). One concern is that given that the authors’ motivation and the two papers’ motivation are so similar – all of the approaches (the two papers’ as well as the authors’) propose to downweight unlikely / outlier tokens, I would appreciate some deeper explanation on the pros and cons of each approach, or why the authors’ approach excels.

---

Update: I read through the paper again, and responded to the authors' rebuttal below -- I don't really have complaints otherwise. I think the motivation and the execution are great.

**Summary Of The Paper:**

Using MLE (maximum likelihood estimation) to train text generation models distributes probability mass to all samples in the dataset, even if the samples are of poor quality or outliers. The authors claim that MLE-trained models would would overestimate the probability of corrupted sequences (which the authors call the “void” of the real data distribution) which leads to degeneration.

To address this problem, the authors propose using total variation distance as shown in Equation (2a) and Equation (2b). Total variation distance would make the model place small or zero probability to low-quality sequences in the training set. However, it’s not easy to apply TVD directly onto text generation, so the authors proposed an upper bound on the TVD (sequence-level) in Proposition 1 / Equation 5. Additionally, given that we do not have the data/reference probability distribution, the target is estimated using a proxy distribution.


**Summary Of The Review:**

Important problem. Hoping for better analysis of the pros and cons of related approaches.

---

> ### Author Response · Authors · 2022-11-16
> **Response to Reviewer PwuP**
>
> We thank the reviewer for the detailed feedback and valuable suggestions to further improve our work.
>
> > 1. All of the approaches (the two papers’ as well as the authors’) propose to downweight unlikely / outlier tokens, I would appreciate some deeper explanation on the pros and cons of each approach, or why the authors’ approach excels.
>
> We first briefly review loss truncation [1] and GOLD [2] and then compare them to our method.
>
> **Loss truncation**
>
> Loss truncation abandons samples with high log loss (also known as the negative log-likelihood) in training. In Proposition 1 of the original paper, the authors prove that the model's log loss on the truncated data distribution with an extra constant sets an upper bound on the total variation distance (TVD). The authors thus propose to optimize on a "simpler" subset of the full dataset in order to achieve smaller log-loss that would tighten the upper bound (note that the $c$-related constant would be a trade-off to the bound as removing more data leads to a larger constant). To achieve this, the authors propose to first heuristically drop the "hard" samples with the highest log losses (hot-start stage), and then train the model on the remaining data (training stage).
>
> Loss truncation can be regarded as downweighting samples at the sequence level with binary weights. However, performing sequence-level sample dropping may systematically lose some specific modes in the data, which is undesirable. For example, we observed an overly short generation length of loss truncation (384 compared to 511 of MLE) on the WritingPrompts dataset as a result of dropping long samples with high log losses. Moreover, determining which samples to drop heavily relies on the first hot-start stage where the model should not be too random or too certain. In practice, it is hard to determine the degree to which the hot-start training should proceed. In our work, we derived our training objective TaiLr from the practical upper bounds of TVD, which softly weights the log loss on the token level. We also analyzed the bias and variance tradeoff in estimating the bound. By appropriately setting the tradeoff hyperparameter, we can reduce the estimation variance at the start of training and gradually decrease the estimation bias as the model becomes more accurate, which circumvents the need of hot-starting the model.
>
> **GOLD**
>
> GOLD learns the generation policy in the offline setting, i.e., calculating rewards on target trajectories by reweighting the policy gradient with importance weights. They made several approximations including simplifying the multi-step importance weights into a single step, and simply assuming the per-step target policy to be uniform. Besides the importance weight, they also came up with several reward functions which score the sequence with another generation model trained by MLE. Although starting with a quite different motivation, the final form of the training objective is somewhat similar to ours.
>
> From an empirical view, GOLD assigns soft weights to samples on the token level based on the importance weight and some hand-crafted reward functions. However, one downside (also discussed in the original paper of GOLD) is that the objective only focuses on capturing the high-likelihood patterns in the data and fails when there are a variety of candidates given the same input in the data. This is also reflected in our experimental results in Table 4, where GOLD has the highest repetition rate and the lowest diversity. In our paper, we analyze the situation where the target distribution has high entropy, and decompose the error of estimating TVD into the bias and the variance term (Section 3.3). We find that the GOLD-style importance weight actually leads to an unbiased but high-variance estimator of TVD, which renders the optimization hard to proceed. In our TaiLr objective, we propose to balance the bias and variance with a hyperparameter, which can be effectively applied to different generation tasks like machine translation, summarization, and long text generation by tuning the hyperparameter.
>
> We summarize the above points and add a discussion of these two models in Appendix C.
>
> > 2. The code is attached but I don't see README.
>
> We update the README in the supplementary material and we will publicly release our code.
>
> [1] Improved Natural Language Generation via Loss Truncation. ACL 2020
>
> [2] Text Generation by Learning from Demonstrations. ICLR 2021

---

> ### Comment · Reviewer_PwuP · 2022-11-29
> **Response to authors**
>
> Thank you, authors, for the response. I have two comments.
>
> First, if there's time, I wonder how the other variants in the GOLD paper would perform. GOLD-s (using token-level rewards instead of sequence-level rewards) seems to perform better for machine translation, summarization, and question generation in the GOLD paper. Or perhaps authors could explain more clearly why the comparison is not needed.
>
> Second, it will be great if the authors define "diversity" more clearly at the beginning of the paper.
>
> I think this is impressive motivation and work. The response makes sense, and I am raising the score.

---

> > ### Author Response · Authors · 2022-12-01
> > **Updated response**
> >
> > Thank you for the encouraging feedback!
> >
> > For the first comment, the other two variants of GOLD, i.e., GOLD-s and GOLD-p, both rely on another generation model trained with MLE to compute the reward of the trajectory (see Equation (6) and (7) in [1]). The only difference is that GOLD-s adds up the probability at each step as the reward while GOLD-p uses the log probability. These two variants introduce additional model parameters in their objective, which needs to load the trained MLE model during training. We believe that it is unfair for our model and other baselines to compare.
> >
> > For the second comment, we define "diversity" in our paper as the coverage of modes in the data distribution, which can be measured by NLL loss or perplexity. This definition is also mentioned by [1]. Although our main goal is to improve the overall generation quality, it is meaningless if the model can only produce one mode in the data [2]. Thus, diversity acts as a constraint for the model to appropriately allocate probability masses to different modes in the data. We will update our paper in the next version.
> >
> > [1] Text Generation by Learning from Demonstrations. ICLR 2021
> >
> > [2] Language GANs Falling Short. ICLR 2020

---

### Official Review · Reviewer_eDTd · 2022-10-25

**Confidence:** 3
**Correctness:** 4
**Technical Novelty And Significance:** 3
**Empirical Novelty And Significance:** 3
**Recommendation:** 10

**Clarity, Quality, Novelty And Reproducibility:**

The paper does a good job in terms of clarity and novelty. Regarding reproducibility, I did not miss any critical experimental detail, although given the complexity of the setup it is virtually impossible to cover all details in the paper, and it would be very valuable to open source the code.

**Strength And Weaknesses:**

STRENGTHS
- This is overall a well-rounded paper: it tackles a very important problem, the proposed solution is sound and well-motivated (although I did not carefully check the correctness of the maths), the paper has more than enough substance, and the reported results are solid.
- Despite being somewhat math-heavy, the paper is written in a very clear and didactic way. I found it easy to follow and an enjoyable read overall!
- Comprehensive (although not entirely convincing, see below) experiments covering 3 downstream applications and a synthetic task, with a good number of baselines.

WEAKNESSES
- The experiments are comprehensive, but limited to rather artificial (or at least not the most relevant) scenarios from today’s perspective. For instance, the authors use a 1-layer LSTM trained on 10k sentences as their oracle for the synthetic data experiments. This sounds ridiculous in 2022, when there are dozens of strong language models publicly available. Similarly, the choice of IWSLT14 as the evaluation benchmark for MT and Gigaword for summarization do not seem the most relevant nowadays (e.g. why not run MT experiments in more languages and/or a more recent WMT benchmark)? In addition, all real-data experiments rely on seq2seq models, while it is decoder-only models that are becoming more and more central in NLP.
- Connected to the previous point, the authors have a decent number of baselines that they implement, but they do not compare to any number previously reported in the literature. If one checks previously published papers, it seems clear that the reported numbers are rather far from the state-of-the-art. For instance, https://aclanthology.org/2021.emnlp-main.534.pdf reports 38.6 BLEU in IWSLT14, compared to 35.1 for the best system in this paper.
- Given the previous points, I think that the experiments in the paper are valid as a proof of concept, but are far from convincing me that I should use the proposed approach when I train my next NLG model. You have convinced me that the approach has some potential, but not necessarily that it is better than standard MLE in the real world.

**Summary Of The Paper:**

This paper explores the Total Variation Distance (TVD) as an alternative to KL divergence for natural language generation. KL divergence is implicitly optimized by the widely used MLE, but the authors argue that this objective is sensitive to noisy data and results in overestimating the probability of corrupted text, while TVD is more robust to outliers. However, one cannot directly optimize TVD, so the authors develop practical bounds and introduce the TaiLr objective that balances the bias-variance tradeoff of estimating it. Experiments on synthetic data, machine translation, text summarization and text generation confirm the effectiveness of the proposed approach.

**Summary Of The Review:**

This is overall a well-rounded paper. It is very well written and has more than enough novelty and substance. The main weakness lies in the empirical side, which is comprehensive and serves as a proof of concept, but falls short in convincing the reader that they should use the proposed approach when training their next model in the real world.

---

> ### Author Response · Authors · 2022-11-16
> **Response to Reviewer eDTd (Part 1)**
>
> We thank the reviewer for the detailed feedback and valuable suggestions to further improve our work.
>
> > 1. "The authors use a 1-layer LSTM trained on 10k sentences as their oracle for the synthetic data experiments. This sounds ridiculous in 2022, when there are dozens of strong language models publicly available."
>
> The choice of the oracle model in the synthetic experiment follows the widely used setting in the language generation literature (especially GANs) [1,2,3], in which the main point is to simulate the sequence structure in the real world with known data distribution and analyze the distributional property of the learned model. Note that to make the setting more realistic, we train the oracle model on real human texts instead of randomly initializing it as previous works did.
>
> We agree that using a more powerful oracle model would make the conclusion more convincing. Thus, we use GPT2-medium as a stronger oracle model and randomly sample 500K sequences with a maximum length of 64 tokens, which include 400K/100K sequences for training/test, respectively. We train a 6-layer Transformer from scratch using the MLE/TaiLr objective on the synthetic data, respectively. We report PPL_test, PPL_oracle, BLEU-4, and SelfBLEU-4 similar to Table 1 in our paper.
>
> | Model              | PPL_test  $\downarrow$ | PPL_oracle $\downarrow$ | BLEU-4 $\uparrow$ | SelfBLEU-4 $\downarrow$ |
> | ------------------ | ---------------------- | ----------------------- | ----------------- | ----------------------- |
> | $p_\textrm{MLE}$   | 138.35                 | 260.09                  | 5.76              | 6.38                    |
> | $P_\textrm{TaiLr}$ | **137.59**             | **253.01**              | **5.96**          | **5.94**                |
>
> The results show that our proposed TaiLr objective consistently outperforms MLE in terms of generation quality (PPL_oracle, BLEU-4) and coverage (PPL_test, SelfBLEU-4) when trained on the samples from a stronger oracle model, i.e., GPT2-medium.
>
> Note that random sampling from GPT2-medium produces a highly diverse corpus that is more difficult for the Transformer model to fit than the LSTM-based oracle model. Therefore, the scale of the perplexity is higher than the numbers reported in the paper.
>
> We update the experimental results in Appendix E.1.1.
>
> > 2. "The choice of IWSLT14 as the evaluation benchmark for MT and Gigaword for summarization do not seem the most relevant nowadays."
>
> IWSLT14 (https://paperswithcode.com/sota/machine-translation-on-iwslt2014-german) and Gigaword (https://paperswithcode.com/sota/text-summarization-on-gigaword) are still among the most widely-used benchmarks to evaluate MT and summarization systems in recent years. As shown in the leaderboards, plenty of recent works still choose to evaluate their models on these two benchmarks. Since we mainly seek to show the generality of our method, we only choose one representative dataset on each task, and may not cover the trending datasets in every specific domain.
>
> As another important reason to choose these two datasets, the main baselines in our paper including GOLD [4], D2GPo [5], and loss truncation [6] have used these two datasets to benchmark their methods. Thus, it is justified for us to compare with them on these datasets.
>
> > 3. "All real-data experiments rely on seq2seq models, while it is decoder-only models that are becoming more and more central in NLP."
>
> In the supervised setting of conditional text generation, seq2seq models (here we mean the model with an encoder and decoder using arbitrary block architectures such as Transformer) are still dominant. On text summarization, the SOTAs are all based on pre-trained encoder-decoder models, e.g., BART [7] and PEGASUS [8] (https://paperswithcode.com/sota/text-summarization-on-gigaword). The long text generation community also discovers that BART outperforms GPT2 when finetuned on long stories [9, 10]. For MT, the top systems in the WMT 2021 shared task [11] are still built based on the Transformer encoder-decoder architecture, e.g., the best model is based on DeltaLM, a pre-trained encoder-decoder Transformer model.
>
> On the other hand, decoder-only models with billions of parameters (such as GPT3) mostly work well in the few-shot/unsupervised setting. However, finetuning these models is expensive, and may even hurt the performance and generalization ability when the model is based on the downstream dataset with a specific domain. To summarize, in this work we propose a new training objective that is used in the supervised learning (or finetuning) setting. Thus, the encoder-decoder model is a more suitable choice than the decoder-only architecture.

---

> > ### Author Response · Authors · 2022-11-16
> > **Response to Reviewer eDTd (Part 2)**
> >
> >
> >
> > > 4. "The authors have a decent number of baselines that they implement, but they do not compare to any number previously reported in the literature. https://aclanthology.org/2021.emnlp-main.534.pdf reports 38.6 BLEU in IWSLT14, compared to 35.1 for the best system in this paper."
> >
> > First of all, since the main goal of this work is to propose a new training objective to remedy the deficiencies of MLE, the baselines we choose are representative works in line with our motivation that modify MLE in different aspects (see Appendix B Additional Related Work).
> >
> > Second, the recent paper (BiBERT, [12]) mentioned by the reviewer focuses on the utilization of pre-trained representations where the authors propose stochastic layer selection of BERT as contextualized word embeddings of the MT model and dual-directional translation. Our contribution is orthogonal to their modifications in model design as we propose a new training objective that is agnostic to the model structure. Our training objective should further boost their performance upon them.
> >
> > To demonstrate this, we replace the CrossEntropyLoss in their codes (https://github.com/fe1ixxu/BiBERT) with our TaiLr objective and train a new model, BiBERT + TaiLr. We keep other hyperparameters verbatim except that we double the batch size in order to run the model on two times fewer GPUs than the code required. We also report the result of the BiBERT we reproduce.
> >
> > | One-Way Model                     | BLEU      |
> > | --------------------------------- | --------- |
> > | BiBERT (Reported by [12], Table 2) | 37.58     |
> > | BiBERT (Our implementation)       | 38.01     |
> > | BiBERT + TaiLr                    | **39.12** |
> >
> > | Dual-Directional Training + Fine-Tuning | BLEU      |
> > | --------------------------------------- | --------- |
> > | BiBERT (Reported by [12], Table 3)       | 38.61     |
> > | BiBERT (Our implementation)             | 38.73     |
> > | BiBERT + TaiLr                          | **39.23** |
> >
> > As shown in the tables above, using TaiLr improves the performance of the models in [12] by 0.5 ~ 1.0 BLEU score. The results demonstrate that our proposed method can be equipped with SOTA models to further boost performance.
> >
> > We update the experiment results in Appendix E.2.1.
> >
> > [1] SeqGAN: Sequence Generative Adversarial Nets with Policy Gradient. AAAI 2017
> >
> > [2] Long text generation via adversarial training with leaked information. AAAI 2018
> >
> > [3] Improving Maximum Likelihood Training for Text Generation with Density Ratio Estimation. AISTATS 2020
> >
> > [4] Text Generation by Learning from Demonstrations. ICLR 2021
> >
> > [5] Data-dependent gaussian prior objective for language generation. ICLR 2020
> >
> > [6] Improved Natural Language Generation via Loss Truncation. ACL 2020
> >
> > [7] BART: Denoising Sequence-to-Sequence Pre-training for Natural Language Generation, Translation, and Comprehension. ACL 2020
> >
> > [8] PEGASUS: Pre-training with Extracted Gap-sentences for Abstractive Summarization. ICML 2020
> >
> > [9] Long Text Generation by Modeling Sentence-Level and Discourse-Level Coherence. ACL 2021
> >
> > [10] Progressive Generation of Long Text with Pretrained Language Models. NAACL 2021
> >
> > [11] Findings of the WMT 2021 Shared Task on Large-Scale Multilingual Machine Translation. EMNLP 2021
> >
> > [12] BERT, MBERT, or BIBERT? A Study on Contextualized Embeddings for Neural Machine Translation. EMNLP 2021

---

> > > ### Comment · Reviewer_eDTd · 2022-11-28
> > > **Update**
> > >
> > > I would like to thank the authors for their detailed response. My main concerns have been addressed in a satisfying manner (including a new SOTA on IWSLT14), and I am thus increasing my score.

---

### Decision · Program_Chairs · 2023-01-20

**Decision:**

Accept: notable-top-5%

**Justification For Why Not Higher Score:**

N/A

**Justification For Why Not Lower Score:**

The paper makes strong theoretical and practical contribution that should be of wide interest in the ICLR community given the current prominence of language models.

**Metareview: Summary, Strengths And Weaknesses:**

The paper analyses Total Variation Distance (TVD) as an alternative to KL divergence and propose a practical training objective for language modelling training based on TVD that is more robust to outliers, down-weighting the probability of low-probability sequences which models trained with MLE tend to overestimate. The approach is validated through experiments on synthetic data, machine translation, summarization and text generation, showing improved results on intrinsic and extrinsic generation quality metrics. The paper therefore makes a strong theoretical and practical contribution. One weakness is that the approach isn’t validated on very large language models, however the paper does show that the method can improve the performance on BERT-like models.

**Note From Pc:**

if the above contains the word "oral" or "spotlight" please see: "oral" presentation means -> notable-top-5% and "spotlight" means -> notable-top-25%. As stated in our emails, we are disassociating presentation type from AC recommendations